# Solo vs. Chorus: Monomers and Oligomers of Arrestin Proteins

**DOI:** 10.3390/ijms23137253

**Published:** 2022-06-29

**Authors:** Vsevolod V. Gurevich, Eugenia V. Gurevich

**Affiliations:** Department of Pharmacology, Vanderbilt University, Nashville, TN 27232, USA; eugenia.gurevich@vanderbilt.edu

**Keywords:** arrestin, oligomerization, signaling, conformation

## Abstract

Three out of four subtypes of arrestin proteins expressed in mammals self-associate, each forming oligomers of a distinct kind. Monomers and oligomers have different subcellular localization and distinct biological functions. Here we summarize existing evidence regarding arrestin oligomerization and discuss specific functions of monomeric and oligomeric forms, although too few of the latter are known. The data on arrestins highlight biological importance of oligomerization of signaling proteins. Distinct modes of oligomerization might be an important contributing factor to the functional differences among highly homologous members of the arrestin protein family.

## 1. Arrestins: A Small Family with Many Functions

Mammals express only four arrestin subtypes [1]. Visual arrestin (a.k.a. S-antigen, 48 kDa protein, and rod arrestin; systematic name arrestin-1) is expressed in rod and cone photoreceptor cells in the retina, whereas arrestin-4 is expressed exclusively in cones at a much lower level than arrestin-1 [2]. Both quench G protein-mediated signaling of photopigments [2,3]. The first family member discovered, arrestin-1 [4], ensures rapid recovery in rods [5,6,7] by shutting down light-induced signaling of the prototypical G protein-coupled receptor (GPCR) rhodopsin [8,9,10] with sub-second kinetics in mammals [11]. The first non-visual arrestin subtype was also discovered as a protein that turns off G protein-mediated receptor signaling [12]. It was termed β-arrestin (systematic name, arrestin-2) because it preferred β2-adrenergic receptor over rhodopsin, in contrast to arrestin-1 with the opposite preference [12,13]. The second cloned non-visual arrestin subtype also quenched G protein-mediated signaling and preferred β2-adrenergic receptor over rhodopsin. Therefore, it was termed β-arrestin2, with the retroactive renaming of the first one, β-arrestin1 [14]. The same protein was cloned from human thyroid and named hTHY-ARRX [15]. As by that time it became clear that non-visual arrestins bind not only β2-adrenergic receptor, but many other GPCRs, systematic nomenclature was proposed, where the number after “arrestin” indicated the order of cloning without implying anything else, so that β-arrestin2 was termed arrestin-3 [16]. The cone photoreceptor-specific arrestin was cloned later and termed X-arrestin in one study [17] and cone arrestin in another [18]. As it was cloned last, its systematic name is arrestin-4. Below, we use systematic names of arrestin proteins.

In addition to precluding GPCR coupling to cognate G proteins, both non-visual subtypes were shown to be involved in various branches of cellular signaling (reviewed in [19,20]), interacting with >100 partners each [21], i.e., demonstrating a remarkable versatility for an average-sized ~45 kDa protein (reviewed in [22]). The overall structures of arrestin-1 [23,24,25], arrestin-2 [26,27], arrestin-3 [28], and arrestin-4 [29] monomers are remarkably similar. However, all arrestin subtypes, with the sole exception of cone-specific arrestin-4, oligomerize [30], and the oligomers they form are strikingly different.

## 2. Arrestin-1

Arrestin-1 crystallized in two labs under different conditions, forming virtually identical tetramers [23,24] (Figure 1A,B), which suggested that it might oligomerize in vivo. Indeed, it was found to exist in a monomer–dimer–tetramer equilibrium in solution [31]. In the crystal tetramer, many elements of arrestin-1 known at the time to be involved in receptor binding were shielded by sister protomers. Thus, an idea that oligomers are storage forms was proposed [31]. Subsequent studies using pulse EPR technique double electron–electron resonance (DEER) showed that the shape of the tetramer of arrestin-1 in solution is very different from that of the crystal tetramer: the four protomers form a closed diamond [32] (Figure 2). This model of the solution tetramer was confirmed by successful disruption of arrestin-1 self-association with mutations targeting predicted protomer–protomer interaction sites [33,34]. In the solution tetramer and both possible dimers forming it, the receptor-binding elements of the monomers were also shielded by sister protomers. Moreover, it was directly demonstrated that only monomeric arrestin-1 binds rhodopsin [35], supporting the idea that if rhodopsin binding is its only function, oligomers must serve some other purpose. The study using multi-angle light scattering (MALS) confirmed that bovine arrestin-1 oligomerizes in solution, forming dimers that associate into tetramers [30]. Measurements of dimerization and tetramerization constants suggested that in case of bovine arrestin-1, tetramer formation is a cooperative process [30]. Monomer–dimer–tetramer equilibrium in solution was found to be a common feature of bovine, mouse, and human arrestin-1, although dimerization and tetramerization constants in these species are quite different [33]. Rods express enormous amounts of arrestin-1 (intracellular concentrations were estimated at ~2 mM), which is the second most abundant protein in these photoreceptors after rhodopsin [36,37,38]. In fact, the level of arrestin-1 in rods is 4–5 orders of magnitude greater than usual levels of the non-visual subtypes in “normal” cells, including neurons, which is in most cases ~20–200 nM [39,40]. Comparison of oligomerization constants with calculated concentrations of mouse arrestin-1 in photoreceptors suggested that the bulk of arrestin-1 in cells must be oligomeric. Apparently, the majority of arrestin-1 molecules in rods exist as dimers or tetramers, depending on the species [33]. The most parsimonious explanation of this is that oligomers are storage forms for arrestin-1 when it is free and not bound to rhodopsin.

However, it has not been clear why arrestin-1 needs to have these storage forms, i.e., why it could not be stored as a monomer. The first indication that the monomer might be harmful for rods came from in vivo experiments in mice using “enhanced” arrestin-1 mutant that binds unphosphorylated light-activated rhodopsin much better than wild type (WT) protein. Arrestin-1 binds with high affinity to active phosphorylated rhodopsin, shutting off the rhodopsin response to light and initiating a rapid recovery process (reviewed in [41]). Defects in rhodopsin phosphorylation or a lack of functional arrestin-1 in humans result in the loss of the rod function and night blindness [42,43]. Mice lacking rhodopsin kinase, which phosphorylates rhodopsin in rods upon light activation, or mice lacking arrestin-1 are effectively blind under normal light conditions, because rhodopsin signaling is not stopped, and therefore rods have no chance to recover [3,44]. We attempted to compensate for the lack of rhodopsin phosphorylation in rhodopsin kinase knockout mice by expressing a phosphorylation-independent arrestin-1 mutant in rods, which was expected to bind light-activated unphosphorylated rhodopsin and facilitate recovery. This compensational approach worked in principle, significantly reducing the recovery time [45,46]. However, the enhanced mutant used, which also happened to be partially oligomerization-deficient [34], when expressed either on the rhodopsin kinase or arrestin-1 knockout background, caused rapid degeneration of photoreceptor cells, the severity of which increased with its expression level [47]. Interestingly, consistent with the improved rhodopsin shutoff kinetics, the retinal degeneration caused by this mutant was light-independent. We have shown earlier that even very high supra-physiological levels of normally oligomerizing WT arrestin-1 were harmless for rods [36]. Furthermore, WT arrestin-1 co-expressed with the mutant dose-dependently protected, albeit partially, the rods from the mutant-induced degeneration [47]. These data suggested that the retinal degeneration caused by the arrestin-1 mutant might be due to an elevated concentration of a cytotoxic monomer. The protection afforded by WT arrestin-1 could then be easily explained by the known ability of oligomerization-competent WT arrestin-1 to draw partially oligomerization-deficient mutants into oligomers [47].

The oligomerization defect was not the only functional characteristic of the mutant different from that of WT arrestin-1. Another feature of the mutant, an enhanced interaction with clathrin adaptor AP2, was earlier shown to be harmful for the photoreceptor cells [48]. Thus, testing whether defective self-association per se is the culprit, required the replacement in vivo of WT arrestin-1 with an oligomerization-deficient mutant that otherwise has WT-like functional characteristics. An arrestin-1 with mutations that selectively impair oligomerization but do not affect rhodopsin binding was constructed and transgenically expressed in arrestin-1 knockout mice, replacing the WT protein [49]. A thorough analysis of these animals showed that non-oligomerizing protein normally quenches rhodopsin signaling [49], consistent with previous finding that arrestin-1 binds rhodopsin as a monomer [35]. However, arrestin-1 mutant with defective oligomerization and WT-like rhodopsin-binding characteristics caused progressive retinal degeneration, which was faster in lines with higher expression and did not depend on illumination at all, i.e., it proceeded at the same pace in dark-reared animals [49]. These data support the earlier conclusion that arrestin-1 monomer at high concentration is toxic for rod photoreceptors. Since rods do express arrestin-1 at very high levels [36,37,38], oligomerization might be required to prevent cytotoxicity. Calculations based on absolute arrestin-1 concentrations in mouse rods [36] and dimerization and tetramerization constants of mouse arrestin-1 [33] suggest that monomer concentration in WT rods is ~ 95 μM [49]. The data with mice expressing oligomerization-deficient arrestin-1 show that rods tolerate up to 500 μM of monomer, but higher concentrations are harmful [49]. Thus, available data suggest that monomeric and oligomeric forms of arrestin-1 are functionally different. The molecular mechanism whereby monomeric arrestin-1 exerts its cytotoxic effect still needs to be elucidated. It is also unknown whether arrestin-1 oligomers (dimers or tetramers) have specific functions in rods that the monomer cannot serve. These functions might be species-specific, as the predominant form of bovine and mouse arrestin-1 in the dark is likely a tetramer, whereas in humans dimers predominate [33].

In the dark, WT arrestin-1 is mostly localized to the rod inner segments and partially to the cell bodies and synaptic terminals, but in the light the bulk of it translocates to the rhodopsin-containing outer segments (OS) [50]. The arrestin-1 translocation has been shown to be a passive diffusion-driven process, with arrestin-1 moving along the gradients produced by its binding to the preferred partners in the light and dark [50,51]. When rhodopsin is activated by photons of light, it becomes rapidly phosphorylated by rhodopsin kinase. Then arrestin-1 monomer binds with high affinity to light-activated phosphorylated rhodopsin [35]. This shifts the monomer–dimer–tetramer equilibrium towards the monomer. As free arrestin-1 in the OS is depleted by its binding to rhodopsin, this process creates a concentration gradient driving the diffusion of arrestin-1 to the OS. In the dark, when rhodopsin is inactive, arrestin-1 is kept in other compartments via its binding to non-rhodopsin partners, most likely to microtubules particularly abundant in the inner segments [52]. Indeed, in the absence of higher affinity partners, such as light activated rhodopsin, arrestin-1 distribution resembles that of microtubules, and this is observed in the case of both WT arrestin-1 [50] and its oligomerization-deficient mutant [49].

Recently, an interesting model was proposed to explain the localization of arrestin-1 in the dark away from its “place of employment”, i.e., OS: oligomers were hypothesized to be too big to fit in the spaces between the rhodopsin-containing discs [53]. Indeed, the rod OS, where rhodopsin and the rest of the signal transduction machinery is localized, are filled with closely packed membranous discs housing rhodopsin with fairly narrow cytoplasmic spaces between them. This model predicts that in the dark, oligomerization-deficient arrestin-1 would be distributed in rod photoreceptors more evenly, with a significantly greater proportion localized to the OS. However, the subcellular distribution of an oligomerization-deficient mutant, which predominantly exists as a monomer, in both dark and light faithfully recapitulated that of the WT arrestin-1 [49], burying this beautifully simple model. The binding of arrestin-1 to the microtubules as a factor holding it in the dark in the inner segments and cell bodies remains the most likely explanation, particularly since both monomeric and oligomeric arrestin-1 bind microtubules well [35,54]. It is worth noting that while arrestin-1 has a microtubule-binding site overlapping with the rhodopsin footprint [54], which is shielded in the oligomeric forms [32], relatively low-affinity binding to microtubules appears to also be supported by the surfaces exposed in oligomers, as microtubule interaction, unlike rhodopsin binding, does not promote the dissociation of arrestin-1 oligomers into monomers [35].

It should be noted that the arrestin-1 translocation process is unlikely to play a role in the normal rod responsiveness to light. First, the light levels that trigger arrestin-1 movement to the OS significantly exceed the rod functional range [37,38,50,55]. Second, translocation of the bulk of arrestin-1 takes several minutes [49,50,55], whereas rods respond to light with sub-second kinetics [11]. Rods are designed to work in dim light (hence night blindness resulting from the rod malfunctions [42,43]), and their sensitivity to light is at the single photon level [56,57]. In dim light, only very few rhodopsin molecules become activated. The data suggest that the amount of arrestin-1 available in the OS in the dark is sufficient for the signal shutoff. The bright light causes extensive rhodopsin activation, which eventually leads to rod saturation making them unresponsive to light [58]. At this point, the bulk of arrestin-1 arrives to the rod OS driven by the binding to activated rhodopsin [50]. It appears likely that the shielding by arrestin-1 of the majority of rhodopsin molecules at light levels where rods no longer transmit signals is protective, helping rods to survive through the daytime to function in dim light in the night [41,59]. In addition, arrestin-1 might facilitate rhodopsin dephosphorylation [60], so that when the light becomes dim enough for rods to function as photoreceptors, rhodopsin emerges fully signaling-competent. This protective function of arrestin-1 in bright light could explain the need for such high level of arrestin-1 expression, as well as for its oligomerization when it is stored away in the dark. Interestingly, the recent observations demonstrate that rods are not completely inactivated in the light, with approximately 10% of function remaining [61,62]. This is believed to be a protective measure preventing a complete closure of the rod calcium channels, which is harmful to rods [63]. The ability of the rods to withstand saturation by the bright light is at least partially due to the translocation of transducin, the visual G protein belonging to the Gi subfamily, away from the OS [64,65,66]. This movement of transducin reduces the intensity of the signaling, prevents saturation, and protects the rods. Transducin translocates in a light-dependent manner in the direction opposite to that of arrestin-1: to the OS in the dark and away from the OS in the light [65]. It is conceivable that arrestin-1 translocation to the OS in the light combined with the storage of oligomerized arrestin-1 in the dark in other compartments serves the same purpose: protecting the rods from damage.

## 3. Arrestin-2

Arrestin-2 crystallized as a dimer [26], which suggested that it also might oligomerize. Self-association of arrestin-2 in cells was described in 2005 [67]. Structural studies and extensive mutagenesis showed that arrestin-2 oligomerization involves two binding sites for an abundant cytosolic metabolite inositol-hexakisphosphate (IP_6_), one on the N- and the other on the C-domain, both on the concave (receptor-binding) side [68]. Importantly, the structure of arrestin-2 in crystals soaked with IP_6_ revealed long chains of arrestin-2, where the protomers interacted in an N-to-C-domain fashion, glued together by IP_6_ molecules between concave sides of the two domains of neighboring protomers [68] (Figure 3). This arrangement suggested that an oligomer is unlikely to bind GPCRs, as receptor-binding surfaces of arrestin-2 in it are shielded by neighboring protomers.

Oligomerization of purified arrestin-2 in solution was demonstrated in 2008 [30]. In contrast to arrestin-1, the arrestin-2 oligomerization was shown to be facilitated by IP_6_ [30], in agreement with in-cell and structural data [68]. The results of the original MALS study of arrestin-2 were satisfactorily explained by the same monomer–dimer–tetramer model of oligomerization developed for arrestin-1 [30]. However, the data collected in a broader range of arrestin-2 concentrations, including much higher ones, did not fit this model, but suggested the formation in the presence of IP_6_ of long chains without a natural limit [69], similar to those observed in IP_6_-soaked arrestin-2 crystals [68] (Figure 3). DEER measurements using arrestin-2 spin-labeled at different positions confirmed this model and showed that in the oligomeric form, arrestin-2 retains its basal conformation earlier revealed by crystallography [26,27], with the C-tail firmly anchored to the N-domain [69]. This is consistent with the idea that in order to enable GPCR binding arrestin-2 must become monomeric.

The functional role of arrestin-2 oligomerization remains poorly understood. The arrestin-2 hetero-oligomerization with arrestin-3, which, unlike arrestin-2, has a nuclear export signal in its C-terminus [70,71], dramatically changed subcellular distribution of arrestin-2: by itself it was present in the cytoplasm and the nucleus, whereas upon co-expression with arrestin-3, it became predominantly cytosolic [67]. Another study suggested that while monomeric arrestin-2 enters the nucleus, its homo-oligomeric form remains largely cytosolic [68], which suggested that both homo- and hetero-oligomerization of arrestin-2 result in the same change of subcellular localization. Thus, oligomers might serve as a storage form, precluding both GPCR binding and translocation to the nucleus. However, considering relatively low intracellular levels of arrestin-2 [39,40], in sharp contrast to arrestin-1 [36,37,38], it is not clear why cells would need to keep it in the oligomeric form. Unfortunately, available data do not suggest any specific function for arrestin-2 oligomers, except a change in subcellular localization.

## 4. Arrestin-3

Virtually every cell in the body expresses both non-visual subtypes. It has been shown that in the brain, where the non-visual arrestin expression is the highest, arrestin-2 greatly outnumbers arrestin-3 [39,40]. This is likely why arrestin-2 was cloned first, two years before arrestin-3: its mRNA in the brain is much more abundant. The sequence of bovine arrestin-3 is 78% identical to that of arrestin-2 [16]. Mild phenotypes of single non-visual arrestin knockouts combined with embryonic lethality of arrestin-2/3 double knockout in mice [72] suggested that these proteins can take over many biological functions of each other. However, the fact that in vertebrate evolution the duality of non-visual arrestins has been retained for hundreds of millions of years, from bony fish to mammals [1], suggests that these arrestins are not redundant. Indeed, multiple functional differences between the two subtypes have been described. Arrestin-2 and -3 differentially regulate GPCR signaling and trafficking [72]. Arrestin-3 has higher affinity for a number of GPCRs, forming in many cases much longer lived complexes [73]. Arrestin-3 also has higher affinity for clathrin [74]. The two non-visual subtypes appear to bind ERK2 differently [75]. Most strikingly, arrestin-3 facilitates the activation of JNK family kinases, whereas arrestin-2 does not [76,77,78].

Although arrestin-2 and -3 were shown to form heterodimers in cells [67], and oligomerization of both is facilitated by IP_6_ [30], this does not mean that homo-oligomers of these two subtypes are necessarily similar. Solved crystal structure of arrestin-3 with bound IP_6_, where arrestin-3 forms a trimer with each protomer in receptor-bound-like conformation, suggests that arrestin-3 oligomers dramatically differ from those formed by arrestin-2 [79] (Figure 4). First, arrestin-2 forms “infinite” chains, whereas arrestin-3 apparently stops at trimer [79], or possibly hexamer (dimer of trimers) [69]. Second, arrestin-2 in its oligomeric form retains basal conformation [69], whereas arrestin-3 in oligomers formed in the presence of IP_6_ assumes a conformation similar to the conformations of arrestin-1 [80,81] and -2 [82,83,84,85] in complex with their cognate GPCRs.

Arrestin-2 and arrestin-3 have two splice variants each, long and short [16]. The short arrestin-3 splice variant lacking 11 amino acids inserted in the long variant between residues 361 and 362 is the predominant arrestin-3 form in all tissues [1,16]. The long variant of arrestin-2, which is the major form, has an insert of eight amino acids encoded by a separate exon [16]. Interestingly, the inclusion into arrestin-3 of this extra eight-residue loop first described in the long splice variant of bovine arrestin-2 [16], and later found to be present in arrestin-2 and absent in arrestin-3 of numerous mammalian and other vertebrate species [1], prevents trimerization of arrestin-3 [79]. This suggests that the presence this loop in arrestin-2 might determine the mode of its oligomerization [79]. It would be interesting to test whether the short arrestin-2 lacking this loop trimerizes in the presence of IP_6_ like arrestin-3, assuming receptor-bound-like conformation, or oligomerizes in the basal state forming chains like the longer isoform of arrestin-2.

The structure of arrestin-3 trimer with IP_6_, in which arrestin-3 assumes a conformation similar to that of receptor-bound arrestin-1 and -2, suggests that this metabolite might serve as a non-receptor activator [79]. However, this state of arrestin-3 does not appear to be functionally equivalent to the GPCR-bound state. While upon receptor activation, both non-visual arrestins facilitate phosphorylation of ERK1/2 [86], the expression of arrestin-3 in cells, where it can form trimers with IP_6_ in receptor-bound-like conformation, does not result in ERK activation without an input from the receptor [87]. The evidence indicates that, in contrast to the arrestin-dependent activation of ERK, activation of JNK3 by arrestin-3 does not require receptor input [76,77,78,87]. It has been suggested that IP_6_-induced transition of arrestin-3 into receptor bound-like conformation within the trimer might substitute for the GPCR-induced transition, thus allowing for receptor-independent facilitation of JNK3 activation by this subtype [88]. Indeed, arrestin-3 mutants with disabled IP_6_ binding sites in the N- and C-domain did not facilitate JNK3 activation, in sharp contrast to WT arrestin-3 [79]. However, this appears to be a correlation, rather than cause and effect. Both non-visual arrestins have so many functions [22] that any set of mutations likely affects multiple ones. For example, arrestin-3-KNC mutant was designed not to bind GPCRs. It was constructed by replacing 12 key receptor-binding residues with alanines [89]. Yet it turned out that arrestin-3-KNC fails to facilitate JNK3 activation [87]. The mutations disabling IP_6_ binding sites are on the same concave sides of both domains where GPCR-binding residues mutated in KNC are localized [79,89], so the mechanism of the effect might be similar. The notion that IP_6_-induced trimer of arrestin-3 facilitates JNK3 activation is inconsistent with the demonstrated ability of a short 25-residue arrestin-3-derived peptide to facilitate JNK3 activation in vitro and in cells [90]: this peptide does not have most of the elements that bind IP_6_ and mediate arrestin-3 trimerization.

Nonetheless arrestin-3 trimerization in the presence of IP_6_ at concentrations found in cells along with a distinct conformation of arrestin-3 in these trimers [79] suggest that this form of arrestin-3 likely plays a different biological role than its monomer. While available data exclude ERK1/2 and JNK3 activation pathways, oligomerization of arrestin-3, which interacts with more than a hundred non-receptor partners [21], might have an impact on select arrestin-3-dependent signaling branches requiring the arrestin-3 conformation found in the trimer. A comparison of the effects of WT arrestin-3 and its trimerization-deficient mutants already constructed [69,79] on other signaling pathways could shed light on this issue.

## 5. Summary and Unanswered Questions

Overall, sequence homology in the arrestin family remained remarkably high for hundreds of millions years [1]. Yet cone-specific arrestin-4 does not appear to oligomerize at all [30], whereas arrestin-1 of all mammalian species tested demonstrates robust oligomerization [33]. Non-oligomerization of arrestin-4 makes sense biologically: it constitutes only ~2% of the arrestin complement in cone photoreceptors, with the rest being arrestin-1 [2]. Sequence comparison suggests a possible structural basis for different behavior of arrestin-1 and -4. The replacement of two conserved in all arrestin-1 proteins phenylalanines (Phe85 and Phe197 in bovine, Phe86 and Phe198 in mouse) with alanines greatly impairs their oligomerization [33]. Phenylalanines in these positions are absent in the arrestin-4 of all species, replaced by Ser/Val in the first position and Leu/Met in the second. This explains why arrestin-4 does not form oligomers similar to those of arrestin-1: the side chain of Phe is significantly greater than that of Ser, or even Val, Leu, and Met. Another structural reason for impaired self-association of arrestin-4 might be that its C-terminus is much shorter than that of arrestin-1 [1]. It has been shown that detaching the arrestin-1 C-terminus from the body of the molecule significantly impairs its oligomerization [47], but the effect of deletion has not been tested experimentally. The details of the structural role of the C-terminus in arrestin-1 self-association cannot be elucidated based on available data: first, this region was not resolved in classical crystal structures [23,24] and was only partially resolved in the most recent one [25]; second, arrestin-1 tetramers in crystal are different from the ones it forms in solution (Figure 1 and Figure 2) [32].

Sequence comparison also suggests why non-visual subtypes cannot form arrestin-1-like oligomers: arrestin-2 and -3 in all species have Val and Ile, respectively, in the first position, and both have Leu in the second. To determine a specific functional role for the oligomerization of arrestin-2 and -3, one needs mutants where this function is disrupted, and all the others remain unchanged. The mutants of this kind of arrestin-1 were constructed [33] and tested in vivo [49], whereas we do not have this type of mutant forms of non-visual subtypes. Considering how many functions both arrestin-2 and -3 have [22], it is not even certain that the disruption of self-association without affecting other functions of these proteins is feasible.

Virtually all soluble proteins are eventually ubiquitinated and then degraded by proteasomes. Whether oligomerization of arrestin-1, -2, and -3 plays a role in the regulation of their ubiquitination and affects the half-life of these proteins in cells is an open question that deserves special investigation. While oligomerization-deficient arrestin-1 mutant appears to be an adequate tool for these studies, this research direction is not straightforward in case of non-visual subtypes, where mutants of this kind are not available.

Finally, it should be noted that oligomerization is not a unique feature of arrestins. It is common among signaling proteins. Functional differences between members of the same protein family are often explained based on their primary structures. This might be the key underlying factor in some cases. However, observed functional differences might arise due to distinct modes of their oligomerization. Strikingly different oligomerization modes of the two non-visual arrestins with greater than 75% sequence identity show that while studying functional specialization of highly homologous proteins belonging to the same family, the structure of the oligomers they form and their conformation within these oligomers should not be overlooked.

## Figures and Tables

**Figure 1 ijms-23-07253-f001:**
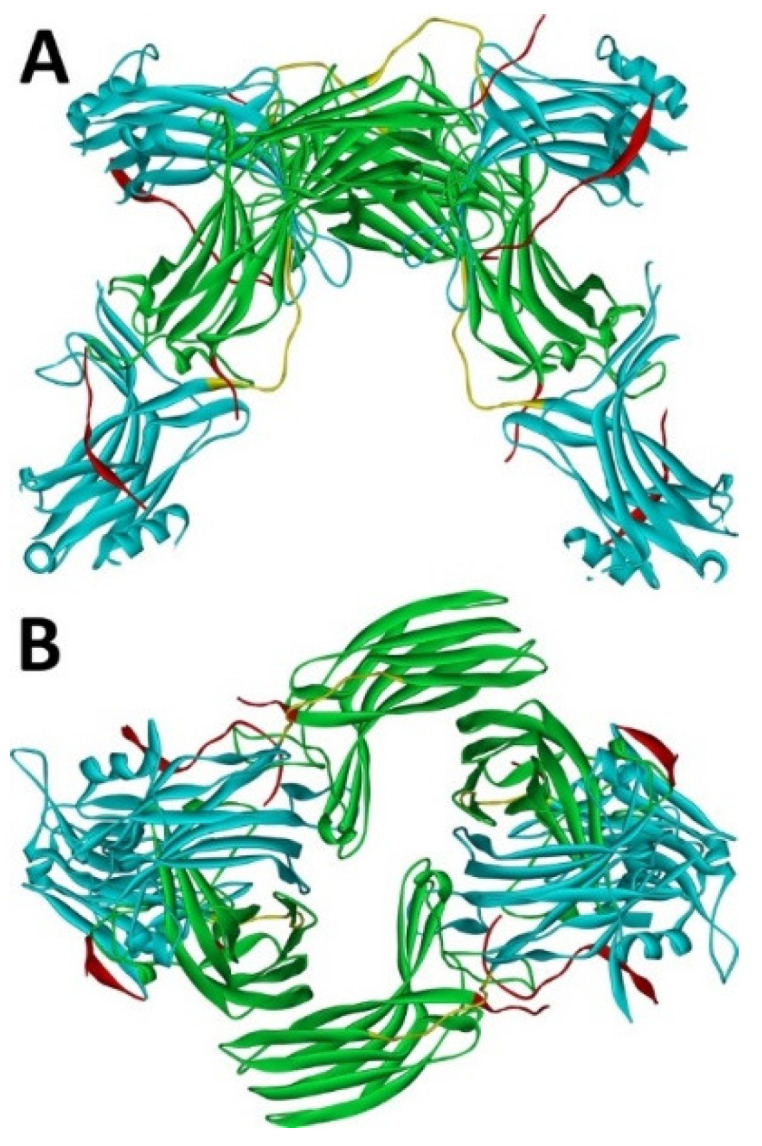
Crystal tetramer of bovine arrestin-1. (**A**), side view. (**B**), top view. PDB ID 1CF1 [24]. In each protomer arrestin-1 elements are colored, as follows: N-domain (residues 1–178), blue; inter-domain hinge (residues 179–190), yellow; C-domain (residues 191–356), green; C-tail (from residue 357; the gap reflects that part of the C-tail are invisible in crystal), red. Images were created in DS ViewerPro 6.0 (Dassault Systèmes, San Diego, CA, USA).

**Figure 2 ijms-23-07253-f002:**
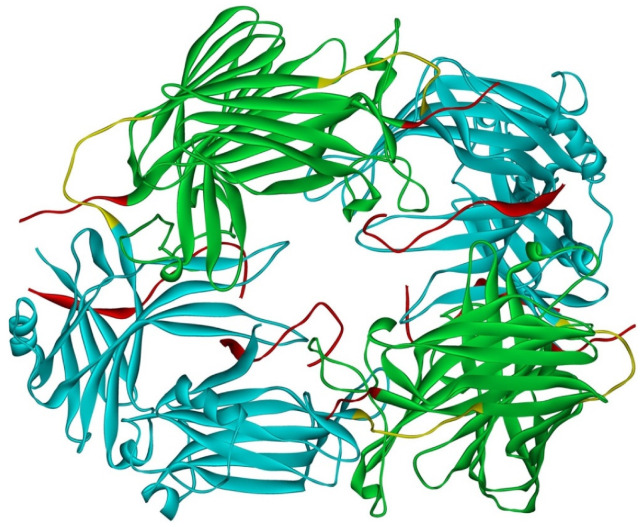
Solution tetramer of bovine arrestin-1. The image is based on the published model [32]. The color code is the same as in Figure 1. Image was created in DS ViewerPro 6.0 (Dassault Systèmes, San Diego, CA, USA).

**Figure 3 ijms-23-07253-f003:**
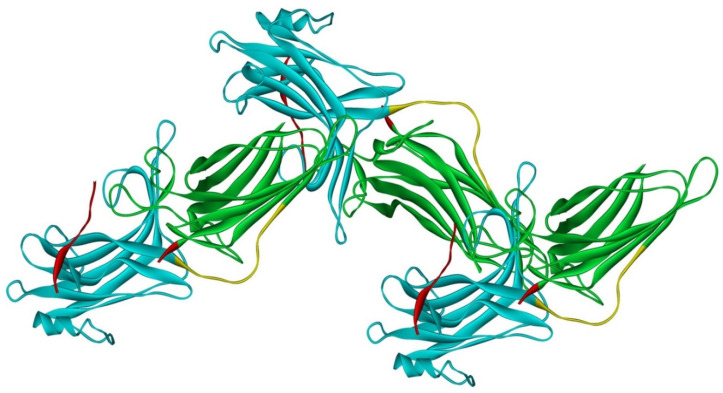
Crystal oligomer of bovine arrestin-2 in the presence of IP_6_. Based on PDB ID 1ZSH [68]. Arrestin-2 oligomer in the presence of IP6 appears to form the same “infinite” chains in solution [69], with each protomer in basal conformation, crystal structure of which was solved earlier (PDB ID 1G4M [26] and 1ZSH [27]). Arrestin-2 in crystal forms “infinite” chains with no apparent limit. Three molecules making IP_6_-mediated N-to-C contacts are shown. In each protomer arrestin-2 elements are colored, as follows: N-domain (residues 1–172), blue; inter-domain hinge (residues 173–184), yellow; C-domain (residues 185–352), green; C-tail (from residue 353), red (note the gap, as part of this element was not resolved in the crystal structure). Image was created in DS ViewerPro 6.0 (Dassault Systèmes, San Diego, CA, USA).

**Figure 4 ijms-23-07253-f004:**
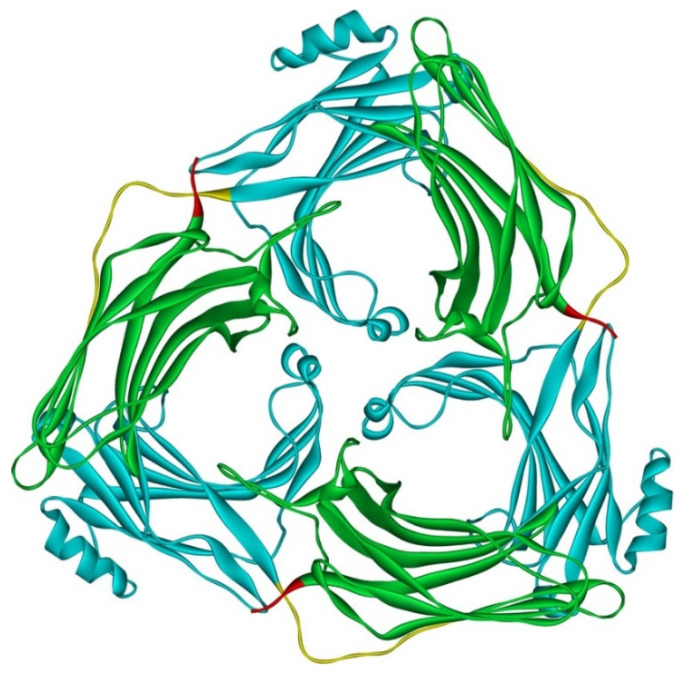
Crystal trimer of bovine arrestin-3 in the presence of IP6. PDB ID 5TV1 [79]. Protomers are linked via N-to-C contacts. Arrestin-3 appears to form the same trimers in the presence of IP_6_ in solution [69]. Each protomer in the trimer is in receptor-bound-like conformation. In each protomer, arrestin-3 elements are colored, as follows: N-domain (residues 1–173), blue; inter-domain hinge (residues 174–185), yellow; C-domain (residues 186–345), green; resolved part of the C-tail (from residue 346), red. Image was created in DS ViewerPro 6.0 (Dassault Systèmes, San Diego, CA, USA).

## Data Availability

Not applicable.

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
