# Peer review of "Solo vs. Chorus: Monomers and Oligomers of Arrestin Proteins"

_ijms, 2022, doi:10.3390/ijms23137253_

Round 1

Reviewer 1 Report

The manuscript is a short review on arrestin oligomerization by two authors that have contributed substantially to the field. The review is of interest to those working in signaling, considering the importance of arrestins in G protein-coupled receptor signaling and internalization. As such, I have only a few minor suggestions for the authors:

a) If I understood correctly, the primary function of oligomerization is to prevent the presence of high concentrations of monomers that seem cytotoxic. Is there any other known or proposed function of oligomerization? Please expand if possible.

b) Why are monomers toxic? One wonders if the processes arrestin monomers interfere with are known or not. Please expand if possible.

c) Are oligomers degraded, or do they have to dissociate into monomers before? Please indicate if relevant information exists.

d) In the version I received, the Greek characters β (beta) and µ (mu) are not shown and were substituted by spaces.

Author Response

Rev 1

The manuscript is a short review on arrestin oligomerization by two authors that have contributed substantially to the field. The review is of interest to those working in signaling, considering the importance of arrestins in G protein-coupled receptor signaling and internalization. As such, I have only a few minor suggestions for the authors:

Thanks!

  1. a) If I understood correctly, the primary function of oligomerization is to prevent the presence of high concentrations of monomers that seem cytotoxic. Is there any other known or proposed function of oligomerization? Please expand if possible.

Thanks for pointing this out. Unfortunately, we still don’t know why arrestin-1 oligomers are toxic for rods. We are working to find that out, as well as to find any other function that arrestin-1 oligomers, but not monomers, might serve. So far, we got no definitive answers. We added sentences regarding this, which are by necessity as vague as the issues.

  1. b) Why are monomers toxic? One wonders if the processes arrestin monomers interfere with are known or not. Please expand if possible.

Thanks for pointing this out. Unfortunately, we still don’t know why arrestin-1 oligomers are toxic for rods. We are working to find that out but do not have definitive answers yet. We added a sentence stating this.

  1. c) Are oligomers degraded, or do they have to dissociate into monomers before? Please indicate if relevant information exists.

This is a very good point we did not think about. While there is no available info on that, this issue certainly deserves investigation. We added brief discussion of this subject in the summary.  

  1. d) In the version I received, the Greek characters β (beta) and µ (mu) are not shown and were substituted by spaces.

Thanks for pointing this out. These errors were apparently introduced in conversion. They are corrected in the revised version (with the hope that they won’t be reintroduced in conversion again).

Reviewer 2 Report

This is a review article regarding the functional diversity of four arrestins in their monomer and oligomer states. This is a narrative review with a large number of references of which many of which are the authors’ own. It is well written and provides a concise picture of the current state of the subject. It should be accepted after fixing minor typos (see below).

Page 1, the first paragraph of the Introduction has lots of typos. In many places, “β” is missing.

L122-123, “~95  M” and “500  M”, unit is missing something (“μ”?).

Author Response

Rev 2

This is a review article regarding the functional diversity of four arrestins in their monomer and oligomer states. This is a narrative review with a large number of references of which many of which are the authors’ own. It is well written and provides a concise picture of the current state of the subject. It should be accepted after fixing minor typos (see below).

Thanks! We would also like to point out that we made an effort to reference all relevant publications. Even though we work on arrestins for 30+ years, the majority (49 out of 90) references are to studies by other labs.

Page 1, the first paragraph of the Introduction has lots of typos. In many places, “β” is missing.

L122-123, “~95  M” and “500  M”, unit is missing something (“μ”?).

Thanks for pointing out both issues. These errors were apparently introduced in conversion. They are corrected in the revised version (with the hope that they won’t be reintroduced in conversion again).